# New Technologies in the Assessment of Carotid Stenosis: Beyond the Color-Doppler Ultrasound—High Frame Rate Vector-Flow and 3D Arterial Analysis Ultrasound

**DOI:** 10.3390/diagnostics13081478

**Published:** 2023-04-19

**Authors:** Emanuele David, Ombretta Martinelli, Patrizia Pacini, Marco Di Serafino, Pintong Huang, Vincenzo Dolcetti, Giovanni Del Gaudio, Richard G. Barr, Maurizio Renda, Giuseppe T. Lucarelli, Luca Di Marzo, Dirk A. Clevert, Carmen Solito, Chiara Di Bella, Vito Cantisani

**Affiliations:** 1Department of Translational and Precision Medicine, Sapienza University of Rome, 00185 Rome, Italy; 2Radiology Unit 1, Department of Medical Surgical Sciences and Advanced Technologies “GF Ingrassia”, University Hospital “Policlinico G. Rodolico”, University of Catania, 95123 Catania, Italy; 3Radiology Unit, Papardo-Hospital, 98158 Messina, Italy; 4Department of Surgery “Paride Stefanini”, Vascular and Endovascular Surgery Division, Sapienza University of Rome, Viale del Policlinico 155, 00161 Rome, Italy; 5Department of Radiological Sciences, Oncology and Pathology, Policlinico Umberto I, Sapienza University of Rome, 00161 Rome, Italymaurizio.renda@uniroma1.it (M.R.); carmen.solito@uniroma1.it (C.S.); chiara.dibella@uniroma1.it (C.D.B.); 6Department of General and Emergency Radiology, “Antonio Cardarelli” Hospital, 80131 Naples, Italy; marcodiserafino@hotmail.it; 7Department of Ultrasound in Medicine, Second Affiliated Hospital, Zhejiang University School of Medicine, Hangzhou 242332, China; 8Department of Radiology, Northeastern Ohio Medical University, Rootstown, OH 44272, USA; 9Southwoods Imaging, Youngstown, OH 44512, USA; 10Interdisciplinary Ultrasound-Center, Department of Radiology, University of Munich, Grosshadern Campus, 81377 Munich, Germany

**Keywords:** atherosclerotic plaque, carotid artery, color-Doppler ultrasound, contrast enhanced ultrasound, Computed Tomography Angiography (CTA), Magnetic Resonance Angiography (MRA), high frame rate vector flow, 3D arterial analysis ultrasound

## Abstract

Atherosclerotic plaque in the carotid artery is the main cause of ischemic stroke, with a high incidence rate among people over 65 years. A timely and precise diagnosis can help to prevent the ischemic event and decide patient management, such as follow up, medical, or surgical treatment. Presently, diagnostic imaging techniques available include color-Doppler ultrasound, as a first evaluation technique, computed tomography angiography, which, however, uses ionizing radiation, magnetic resonance angiography, still not in widespread use, and cerebral angiography, which is an invasively procedure reserved for therapeutically purposes. Contrast-enhanced ultrasound is carving out an important and emerging role which can significantly improve the diagnostic accuracy of an ultrasound. Modern ultrasound technologies, still not universally utilized, are opening new horizons in the arterial pathologies research field. In this paper, the technical development of various carotid artery stenosis diagnostic imaging modalities and their impact on clinical efficacy is thoroughly reviewed.

## 1. Introduction

Atherosclerotic carotid artery plaque is a leading cause of death and ischemic stroke [1]. Worldwide, approximately 21% of people aged 30–79 years have carotid plaque which means that approximately 816 million people suffer from carotid plaque. It is more common in people over 65 years of age and in men than in women [2]. Thromboemboli originating from an ipsilateral carotid stenosis are the cause of a substantial proportion of ischemic strokes. Furthermore, the presence and degree of atherosclerosis in the carotid arterial system are currently used as markers to estimate an individual’s cardiovascular risk.

Accurate early diagnosis of carotid artery stenosis (CAS) is therefore crucial to making clinical decisions on the follow-up treatment profile and for the choice of medical or surgical treatment. The current guidelines have used the degree of carotid stenosis as the main criterion for selecting treatment options for decades. Hemodynamic changes such as the blood flow velocity, flow direction, flow pattern, and functional evaluation are other main factors in evaluating carotid stenosis.

The currently available imaging techniques include color-Doppler ultrasound (CDUS), contrast-enhanced ultrasound (CEUS), computed tomography angiography (CTA), magnetic resonance angiography (MRA), and cerebral angiography (DSA) [3]. The diagnostic sensitivity of carotid stenosis by these imaging modalities varies from 31 to 85%, and their specificity varies from 54 to 85% [4].

Carotid CDUS is certainly the first level examination in the evaluation of carotid artery disease; it provides precise and accurate information about carotid atherosclerotic plaque burden assessment and for compositional analysis. However, the CDUS examination may not be enough for an accurate diagnosis of CAS in the presence of large, irregular or calcified plaques.

An overview of the current imaging modalities for CAS detection is reported, focusing on the advances in US technology such as contrast enhanced ultrasound (CEUS), high frame rate Vector flow (V-flow), and 3D arterial analysis ultrasound (3D-US) that have significantly improved the diagnostic accuracy of US evaluation.

## 2. Methodology Section

The present narrative review is based on the evaluation of the most recent existing literature by means of PubMed and Embase databases. The keyword used were: Atherosclerotic plaque; Carotid artery; Color-Doppler ultrasound; Contrast enhanced ultrasound; Computed Tomography Angiography (CTA) and Magnetic Resonance Angiography (MRA); High frame rate vector flow; 3D arterial analysis ultrasound. The main and most authoritative papers were selected, based on the scientific impact of the journal as well as on case studies, selecting those performed on a wider population. First of all, a flow chart of searching envisaged the search for the main papers, reviews or original papers on vascular field, concerning the imaging of atherosclerotic plaque, and then papers focusing on individual methods such as color -Doppler ultrasound, contrast enhanced ultrasound, computed tomography angiography, and magnetic resonance angiography. Finally came papers dealing with the innovative techniques of High frame rate vector flow and 3D arterial analysis ultrasound. The full text was retrieved and evaluated to assess the features, the scientific research and clinical value, and the limitations of the abovementioned techniques.

## 3. Computed Tomography Angiography (CTA) and Magnetic Resonance Angiography (MRA)

CTA is one of the most commonly used imaging examinations for accurately evaluating the degree of luminal narrowing. It has very high sensitivity (98%), positive predictive value (93%), and inter-operator reliability in evaluating vessel patency and luminal narrowing compared with DSA [5]. One benefit of using CTA for plaque evaluation is the relative standardization of the imaging technique across platforms and institutions, although it is mostly used as a second-level technique, especially in surgical planning or in the first instance in acute stroke.

CTA is generally acquired after the intravenous (IV) administration of non-ionic iodinated contrast, often using bolus-tracking software. Helical mode CTA scanning is then generally performed with a multidetector scanner from the aortic arch to the C1 ring with submillimeter (frequently 0.625 mm) resolution. Post-processing techniques for three-dimensional CTA imaging (e.g., maximum intensity projection, shaded surface display, and volume-rendering techniques) can be performed in a few minutes and provide images comparable with those obtained with DSA [6]. Multiplanar reconstructions are performed to fully evaluate the vessels and to properly account for inherent vessel tortuosity. CTA allows the evaluation of the entire course of the carotid artery from the aortic arch to the intracranial segments, and the stenotic severity at all levels of the vessel (Figure 1).

The main advantage of CTA is the possibility of examining the extra- and intracranial arteries at once, and it basically meets all clinical needs, both for acute stroke treatment, which usually involves intracranial occlusion, and in prevention, which involves CAS. However, it has some limitations; it does not allow hemodynamic evaluation, and uses ionizing radiation and nephrotoxic contrast media. CTA examinations are also less cost-effective and are limited in the evaluation of vessel hemodynamics [7].

MRA represents an alternative second line method which does not use ionizing radiation. Contrast-enhanced MRA (CE-MRA) and Time-of-flight (TOF) MRA are the most frequently used MRA techniques [6]. CE-MRA provides bright lumen signals suitable for measuring stenosis in line with its angiographic effect, due to the passage of a gadolinium contrast bolus through the arteries after IV injection [5,8]. TOF-MRA does not require contrast medium IV injection, and so it is an alternative sequence for patients who have contraindications to their use [5]. Both MRA techniques are helpful for the evaluation of atherosclerotic carotid artery disease, although CE-MRA is more accurate than TOF sequences (Figure 2) [5,9].

MRA carries the advantage of being devoid of radiation, but this technique is fraught due to its expense and because it is time consuming and not readily available. It is also unable to depict calcifications and plaque burden, particularly in light of positive remodeling, nor it is able to detect high-risk components of the atherosclerotic plaque [10,11].

In spite of the use of iodinated contrast agents and the exposure of the patient to ionizing radiation, CTA has several advantages over MRA for carotid imaging. CTA is less susceptible to artifacts than MRA, technical, and based on patients and calcium; other advantages over MRA include that CTA provides more accurate information about the surrounding anatomy, thus being useful in surgical planning, and it is far more widely available than MRI (Table 1) [12].

## 4. Color-Doppler Ultrasound (CDUS)

CDUS using linear probes is a safe, easily available, and powerful technique for visualizing the carotid arteries, with the advantage of coupling the real-time morphological imaging with the evaluation of hemodynamic changes (Figure 3).

There are several methods of US measurement of the degree of arterial stenosis. More precise methods include two-dimensional (2D) quantification techniques such as the North American Symptomatic Carotid Endarterectomy Trial (NASCET) method, which measures stenosis by taking the inner diameter of the distal normal lumen as the basic inner diameter and the inner diameter of residual lumen at stenosis segment as the measurement [13]. 

The European Carotid Surgery Trial (ECST) method compares the diameter of the stenotic area with the normal diameter of the carotid bulb [14].

The Common Carotid method (CC) measures the diameter of the residual lumen at the most stenotic portion of the artery and then compares this to the luminal diameter in the proximal CCA [15]. 

The degree of stenosis may vary according to these different criteria, since the NASCET criteria of 50% stenosis is roughly equal to 75% stenosis by ECST criteria (Figure 4).

The combined peak systolic frequency (PSF), peak systolic velocity (PSV), and end-diastolic velocity (EDV) serve as criteria for the ultrasound assessment of a carotid stenosis degree greater than 70%.

Standard two-dimensional (2D) imaging of arterial plaque depends on the operator’s skills and variable image quality. Other 2D shortcomings include limited planar information of the plaque extent and an improper insonation angle, which can lead to measurement errors of carotid artery stenosis. 

Some pitfalls of a velocity-based evaluation of carotid stenosis are higher flow velocities due to arterial tortuosity or compensatory blood flow via collaterals from contralateral ICA occlusion. In this case, the ratio of flow velocities in the ICA and CC artery (CCA) may help to detect true carotid stenosis. 

An increased flow velocity can also be recorded within a carotid artery stent because of the decrease in vessel wall compliance. Additionally, high carotid bifurcation or extensive wall calcification may reduce the CDUS accuracy. CDUS may also fail to distinguish between sub-occlusion and complete carotid artery occlusion [16,17]. 

Moreover, when the carotid stenosis rate is >90%, the PSV of some patients will decline, which means that the stenosis rate does not match the PSV value [18].

## 5. Contrast Enhanced Ultrasound (CEUS)

Among diagnostic methods, CEUS, which uses a contrast medium based on sulfur hexafluoride, has emerged in the last decade as a reliable technique not only due to its ability to quantify the grade of stenosis, but also for its superior capability in depicting the vulnerability features of the plaque, thus providing an accurate qualitative assessment and stratification of the risk of rupture. It also represents a valid method in the evaluation of carotid dissection. CEUS uses an intra-vascular contrast agent consisting of microbubbles (1–8 µm) filled with perfluorinated gas with low solubility injected to acquire high contrast ultrasonic images of the carotid artery. It allows some limitations of DUS to be overcome, such as the detection of low blood flow and insonation of deep vessels (Figure 5) [12,19,20]. 

Other than the degree of stenosis, the CEUS technique provides reliable information about plaque morphology and plaque composition, which plays an important role in characterizing vulnerable and ulcerated carotid plaques that are important in the assessment and stratification of the stroke risk [21,22,23,24]. CEUS allows for the more accurate delineation of the plaque surface and for the diagnosis of ulceration denoted by a 1 mm × 1 mm microbubbles column within an atherosclerotic lesion. When compared to conventional carotid angiography, this criterion of plaque ulceration by CEUS showed that a 100% sensitivity, specificity, and accuracy was achieved [25,26]. Inflammatory cell infiltration and intraplaque neovascularization are markers for vulnerable plaques. CEUS is a valuable imaging tool to assess intraplaque neovascularization, defined from moderate to extensive regarding the visible appearance of moving bubbles in the plaque from the adventitial side or plaque shoulder to the plaque core. Quantitative software analysis of intraplaque neovascularization on CEUS using specific quantification algorithm is now available for automated quantification of intraplaque micro-vessels. The degree of intraplaque neovascularization as detected by CEUS imaging closely correlates with the histological grade of vulnerability of post-endarterectomy plaques [27]. 

CEUS has also proved to be effective in detecting extra-cranial carotid and vertebral artery dissection, particularly in patients with renal failure who cannot undergo CTA [28]. However, it should be remembered that CEUS is not exempt from the already known limitations of US; it is operator dependent and may not be sufficient in some cases, for example, in the presence of large and calcified plaques (Table 2).

## 6. High Frame Rate Vector Flow (V-Flow) and 3D Arterial Analysis Ultrasound (3D-US)

V-flow represents an emerging quantitative US method to assess the blood flow characteristics in the carotid artery for superficial vessels, mainly focused on the carotid artery [29]. Compared to conventional US, real-time high frame rate V-flow is a new method to measure the wall shear stress (WSS), which is the frictional force exerted on the endothelial surface of the vessel wall and is strongly influenced by hemodynamic changes related to carotid stenosis (Figure 6) [30,31].

V-flow imaging technique can also be combined with contrast-enhanced CEUS to enhance the echogenicity of the blood pool, improving image quality. This method, for which a role is already being carved out in the literature in the evaluation of liver lesions, allows the representation of particle image velocimetry (PIV), enabling two-dimensional blood flow quantification [32,33].

Blood flow is characterized by tracking the displacement of a group of microbubbles on a frame-to-frame basis through cross-correlation analyses. Moreover, it employs unfocussed plane wave transmissions, allowing for high frame rate imaging and consequently the tracking of high velocities and transient flow phenomena [34,35,36]. 

Quantifying blood flow is of high value in the assessment of atherosclerotic plaques, especially unstable ones; this requires blood flow visualization (preferably in three dimensions) with a high spatial and temporal resolution. 

V-flow measurement is a rapid, simple, and feasible imaging technique for the wall shear stress (WSS) assessment of common carotid arteries, which will probably be an important tool for assessing common carotid arteries’ function.

New perspectives are being gained through the 3D technique. While providing a model in three spatial planes, the software provides a read-out of the quantitative analysis of maximum stenosis and plaque volumetric measurement of the plaques; this can be made with a 3D-US system based on 2D-US image acquisition and can be measured accurately and with low variability, making it a useful tool in clinical studies of the progression and regression of carotid plaques. 

An ultrasound of the common carotid artery can lead to prediction and even clinical management of future cardiovascular disease through the measurement of Intima-Media Thickness (IMT), which usually is performed in order to detect atherosclerotic disease. This information can lead to the prediction and even clinical management of future cardiovascular disease. However, the 2D evaluation of Intima Media Thickness shows its limitations when stenosis by an atheroma is present. By using 2D imaging, the angle at which the B-mode image was acquired can affect the measurement. Additionally, for patients that present with irregular atheroma patterns, there is no way of precisely quantifying their conditions. 

Three-dimensional arterial analysis provides a volumetric measurement of the plaques to better evaluate the risk of population with or without known stenosis. This quantification leads to reduced inter-observer variation as it results in dependable measurements at a consistent location. 

Three-dimensional arterial analysis also provides an intuitive visualization of plaque formation, location, shape, and distribution by three-dimensional remodeling. This is a qualitative representation of plaques which complements the quantitative diagnosis and enhances the accuracy of diagnosis.

The 3D arterial analysis sees the application of a directional force to the tissue to cause deformity of the vascular wall.

It is represented in a colorimetric map based on the tissue stiffness, which shows vulnerable areas and stratifies the risk. 

Recent works in the literature confirm 3D-US to have excellent intra- and inter-observer reproducibility and excellent agreement with 2D-US and angiography for the evaluation of carotid disease. Further studies assessing the reliability of carotid plaque characteristics using 3D-US in symptomatic and asymptomatic patients are required [37].

New developments regarding 3D arterial analysis show the combination with Contrast Enhanced Ultrasound (CEUS).

Indeed, Cantisani et al. compared CDUS, 3D-Arterial analysis and CEUS, Computed Tomography angiography (CTA), and histology in the assessment of carotid plaque vulnerability and carotid stenosis degree; CDUS provided lower sensitivity and specificity (respectively, 84.6% and 80%) in the evaluation of plaque stenosis; meanwhile 3D arterial analysis and CEUS obtained a sensitivity of 96.7% and 89%, respectively, with a specificity of 100% highly comparable with CTA, in the evaluation of the stenosis degree. Indeed, according to their paper, 3D arterial analysis and CEUS had a higher diagnostic accuracy than reported in the literature, when compared with CTA used as a reference method. However, they underlined that the daily application of 3D techniques effect the probe physical size and weight [38].

Therefore, it has been postulated that the combination of CEUS and 3D arterial analysis may provide a powerful new clinical tool to identify and stratify “high-risk” patients with atherosclerotic carotid artery disease, identifying vulnerable plaques that need to be treated. These applications may also help in the post-operative assessment of treatment options to manage cardiovascular risks [27]. 

However, new developments and technical refinements are mandatory for this technique to become a routine tool for quantifying carotid disease progression and regression. It is currently limited by the greater size and weight of the transducer compared with 2D array and the need for dedicated quantification software to process a 3D image which takes 1–2 s for a matrix transducer; additional software, which may use artificial intelligence, is advised. Though not yet applicable in clinical practice, 3D vessel wall imaging may also improve the prognostic value of the carotid intima-media thickness testing for the noninvasive assessment of atherosclerosis.

Finally, data for the grading of plaque volume are not standardized and threshold cut-off values must be defined by means of a multicenter study on a larger patient population. Nonetheless, the present evidence suggests that 3D qualitative representation is combined with a more accurate evaluation of plaque morphology, surface, and volume which enhances the accuracy of evaluating the plaque burden and predicting cardiovascular risk. This technique also allows us to follow the carotid plaque over time to monitor the response to treatment (Figure 7) (Table 3).

## 7. Discussion

Over the previous decades, there has been a paradigm shift in the risk stratification of CSA. Not only the degree of luminal narrowing, but also the plaque morphology and composition play an important role in assessing the risk brain ischemia of the carotid plaques. Indeed, according to the most updated professional society guidelines, carotid artery revascularization is recommended to treat asymptomatic carotid artery stenosis of 60–99% only if associated with one or more characteristics that may be associated with an increased risk of late ipsilateral CVA [39,40]. Indeed, it is necessary to identify imaging criteria that might reveal an increased risk of CVA on BMT [41]. Plaque surface irregularity and texture characteristics have proved to be independent predictors of the occurrence of cerebrovascular ischemic events in the general population, independently of the stenosis degree evaluation. Thus, in recent years the atherosclerotic plaque vulnerability features have been extensively evaluated for the CVA risk estimation, especially in asymptomatic individuals who are eligible for medical treatment or revascularization. In fact, ipsilateral ischemic cerebrovascular events are more common in patients with high-risk plaques [42,43]. Ulceration, neovascularization, inflammation, thin fibrous cap, lipid core, and intra-plaque hemorrhage are all recognized causes of plaque vulnerability [44,45]. Neovascularization (IPN) and hemorrhage are histopathological features associated with a vulnerable plaque that, in case of rupture, could lead to a CVA [46,47]. This has led to further advancement in the imaging tools for carotid plaque detection. Imaging has been developed to a high level of sophistication. Improvements from one dimension (1D) to 2D images, and from 2D images to 3D models, have revolutionized the field of imaging. This not only helps in diagnosing various critical and fatal diseases in the early stages, but also contributes to making informed clinical decisions regarding the follow-up treatment profile. The use of computer-aided programs has further improved the sensitivity, specificity and accuracy of CAS diagnosis through various imaging modalities. CEUS is an effective recognized imaging modality vascularization assessment, with many clinical applications (i.e., neoplastic lesion, blunt trauma, inflammation, aortic endoleaks after EVAR, kidney transplant evaluation, kidney lesion characterization, kidney cystic assessment, pediatric lesion characterization, etc.) [48,49]. Since plaque neo-vascularization has been a consistent feature of plaque vulnerability, many researchers have proved the correlation between high risk or vulnerable plaques and contrast enhancement with associated intraplaque neovascularization and inflammatory changes [50,51]. The contrast enhancement grade has been reported to be directly related to increased inflammatory infiltrate, and the late-phase contrast enhancement of plaque has been associated with inflammatory plaque infiltration [52]. Feinstein reported that early phase CEUS could identify plaque neovascularity and provide an enhanced delineation of plaque anatomy, including ulcerations in comparison to gray-scale or color-Doppler US [53]. It is worth noting that SonoVue is a blood pool agent [54]; Hoogi et al. showed an indirect correlation between contrast enhancement and the degree of inflammatory infiltrate [55]. Indeed, the most recent EFSUMB guideline strongly recommends CEUS use in carotid stenosis to better differentiate between total carotid occlusion from carotid sub-occlusion and to identify intraplaque neovascularization well; lesser evidence was reported for dissection evaluation and for inflammatory vascular aortic disease assessment [19]. CTA provides a detailed picture of the carotid arteries. A2D slices, a 3D reconstruction of CTA slices, can be performed for a better visualization of carotid plaque morphology, but it may be inaccurate in evaluating carotid artery stenosis, especially in presence of vessel calcification. Additionally, CTA suffers from X-ray radiation exposure and contrast use drawbacks, with the inherent risks of cancer, allergic reaction, and contrast induced nephropathy. More recently, AI has been applied to CTA. In particular, Dong Z et al. [56] compared the diagnostic performance of Radiomics versus conventional CT carotid artery features to identify symptomatic patients with carotid artery atherosclerosis. A radiomics-based ML model was fitted on the training set, and the radiomics-based ML model and conventional assessment were compared using the area under the curve (AUC) to identify symptomatic participants after excluding participants with other stroke sources. Their preliminary results showed that radiomics-based ML analysis increased the accuracy of carotid CTA in the identification of recent ischemic symptoms in patients with carotid artery atherosclerosis. However, although it seems a promising research field of application, further studies are mandatory to confirm which role AI and new CT technologies and software may have in increasing the accuracy and reduce dose exposure to the patients. MRA is another effective method, with high accuracy and resolution for vulnerable plaques and a degree of carotid artery stenosis assessment. However, MRA is not widely available, it is more expensive than CDUS and CTA, and it cannot be performed in patients with contraindications. In conclusion, CDUS is a low-cost, reliable tool in picking up carotid atherosclerosis which provides real-time morphological imaging and hemodynamic evaluation of carotid artery plaques. However, it requires a trained operator to perform and interpret, and high carotid bifurcation or extensive wall calcification may affect its accuracy. CDUS may also fail, which may help patients to make clinical decisions about further treatment plans combined with a high-frame rate vector flow imaging technique (V-flow) or the other vascular identification techniques, especially if implemented by computer-based diagnosis algorithms, which can potentially address these issues, bringing a substantial improvement to the overall performance of CDUS. Several pieces of evidence have shown that CEUS imaging is a valuable tool to evaluate plaque surface irregularities and ulceration as well as intraplaque neovascularization and inflammation. The 3D-US examinations are also superior to 2D-US examination for the assessment of the whole surface, the echo-structure, and the volume measurement of the plaque. Previous studies have proved that the V-flow can visually and quantitatively evaluate complex flow behavior at a plaque level for assessing the degree of carotid stenosis more accurately than CDF. It follows that the combined use of V-flow and conventional ultrasound has broad application prospects in the diagnosis of severe carotid stenosis. According to current knowledge, wall shear stress (WSS) is thought to play a critical role in the local development of atherosclerotic plaque and to affect plaque vulnerability. Ultrasound vector flow imaging, which provides directional information on velocities and more excellent temporal and spatial resolution, seems to be able to estimate WSS accurately. The 3D analysis method allows the precise characterization of plaque surface irregularity and ulceration and carotid plaque echo-structures with good intra- and inter-observer reproducibility. Besides the stenosis degree, plaque echo-structure and surface irregularity are important in determining the risk of CVDs in carotid artery disease. The important role of carotid artery plaque analysis in stroke risk stratification makes it necessary to increase sophisticated and innovative approaches that must not only be validated but must also translate into clinical practice. Especially in the asymptomatic group, a multiparametric US study with CEUS and 3D arterial analysis in combination provides a better tool to identify and stratify the carotid plaques at high risk of stroke. These applications may also help in the postoperative assessment of treatment options to manage cardiovascular risks. However, progress is still required for these US techniques to become a routine tool for detecting high-risk carotid plaques which predispose patients to an elevated risk of cerebrovascular events in order to make clinical decisions about the best treatment plans. Multicenter randomized studies are mandatory to assess all these new techniques, and future multidisciplinary guidelines are warranted to provide new rules in clinical, diagnostic, and interventional treatments of patients with carotid artery disease.

## Figures and Tables

**Figure 1 diagnostics-13-01478-f001:**
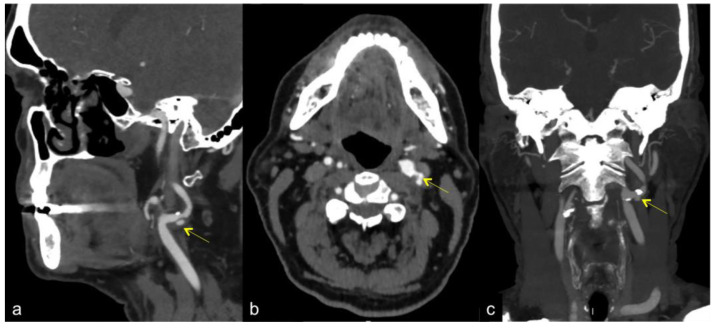
CTA arterial phase sagittal (**a**), axial (**b**), and coronal (**c**) multiplanar reconstructions show an ulcerative left carotid internal artery plaque (arrow; (**a**)–(**c**)).

**Figure 2 diagnostics-13-01478-f002:**
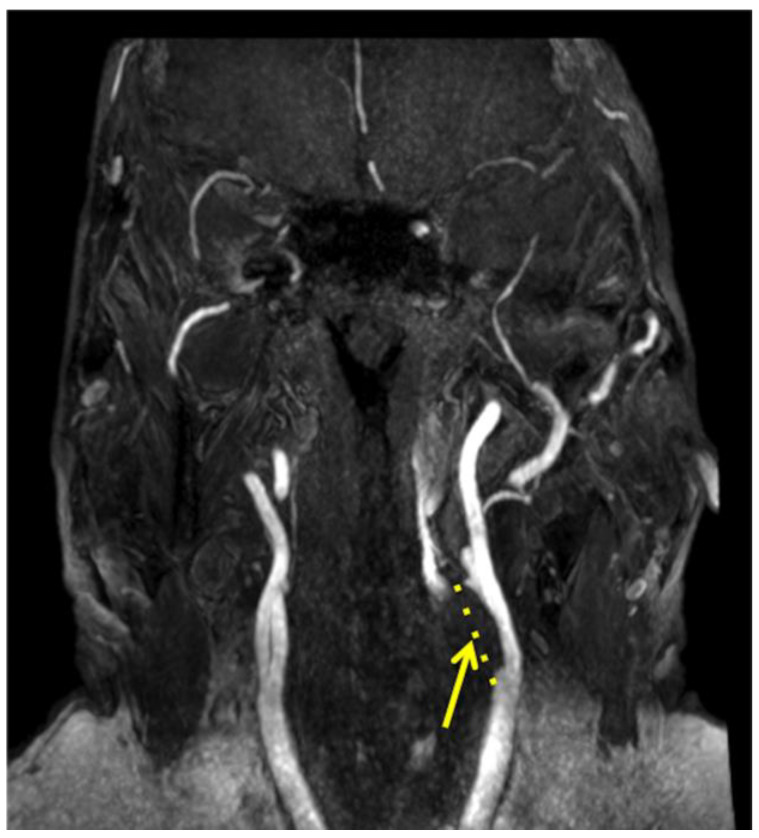
TOF-MRA coronal view shows left partial occlusion of the left common carotid artery at the bulb (arrow) as well as near partial occlusion of the proximal left internal carotid artery (dashed line).

**Figure 3 diagnostics-13-01478-f003:**
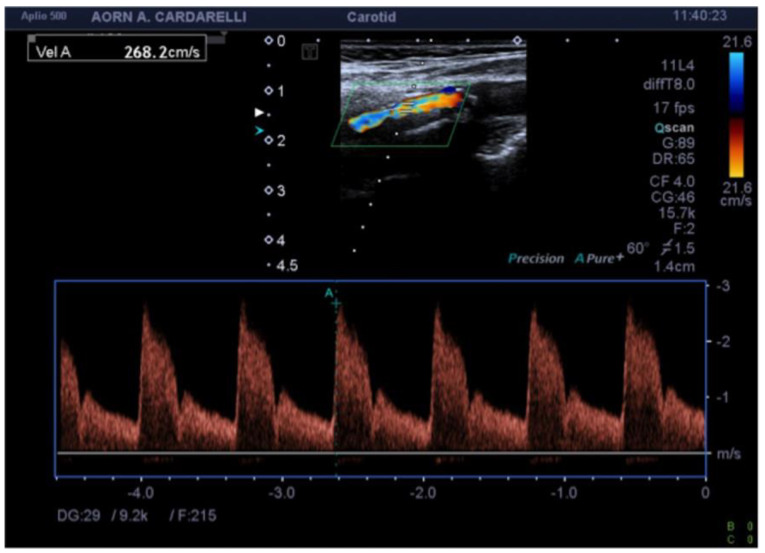
CDUS shows evidence of high grade stenosis (>70%) of a right internal carotid artery stenosis with aliasing at colormap (top) and increased systemic peak velocity to spectral flow analysis (below).

**Figure 4 diagnostics-13-01478-f004:**
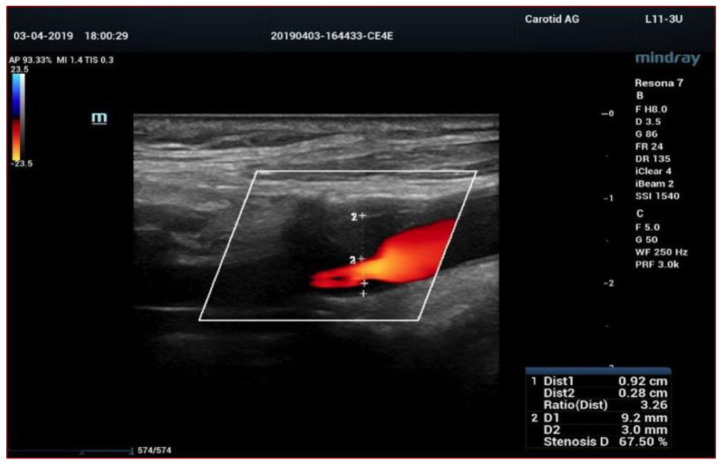
CDUS carotid stenosis measured by ECST method.

**Figure 5 diagnostics-13-01478-f005:**
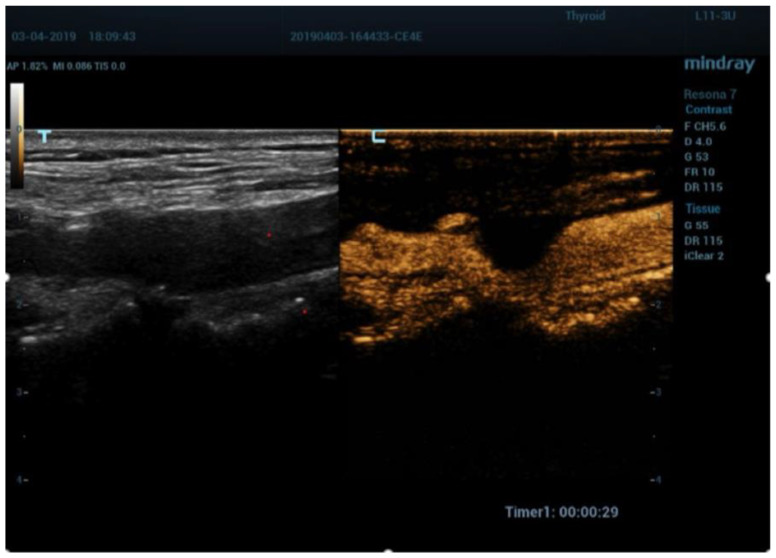
CEUS of carotid stenosis at 29 s arterial phase. Same patient as Figure 4.

**Figure 6 diagnostics-13-01478-f006:**
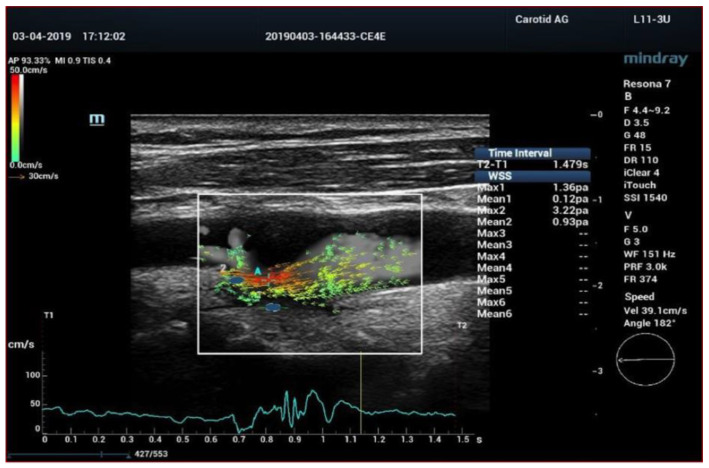
V-flow imaging shows the turbulence and direction of flow with entity estimation of plaque stenosis.

**Figure 7 diagnostics-13-01478-f007:**
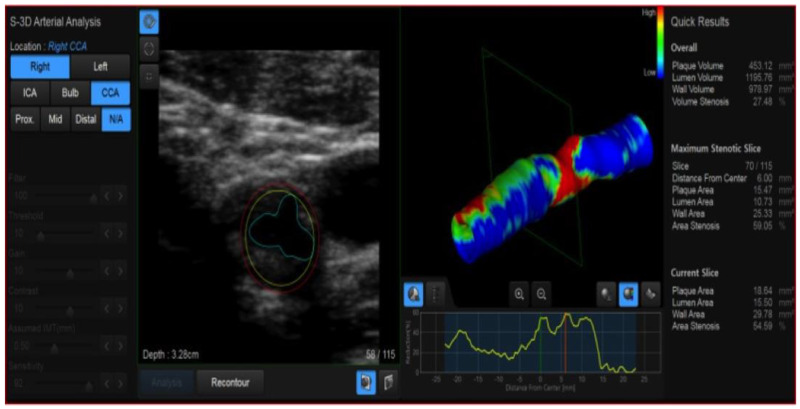
3D-US carotid stenosis evaluation.

**Table 1 diagnostics-13-01478-t001:** Advantages and disadvantages of various vascular imaging modalities. Modified from [5].

	CTA	MRA	CDUS	DSA
Advantages	Widely available Rapid data acquisition Flow independent technique Accuracy close to DSA Fewer motion artifacts	No contrast needed (TOF)	Inexpensive Portable Widely available	High-spatial resolution Immediate treatment Gold standard for therapeutic decision making
Disadvantages	Contrast dependent Radiation exposure No hemodynamic evaluation	Limited availability and feasibility Flow and motion artifacts Inferior accuracy to CTA/DSA	Operator and experience dependent Strongly calcified plaques with large acoustic reverb artifact Low panoramic view	Contrast dependent Radiation exposure Peri- and post-procedural complications Availability still limited

**Table 2 diagnostics-13-01478-t002:** CEUS advantages and disadvantages for the detection of carotid stenosis.

	CEUS
Advantages	Accurate delineation of plaque surface and ulceration Inflammatory cell infiltration and intraplaque neovascularization detection A quantitative software enhancement plaque analysis Alternative to CTA in renal failure patients or iodinate cross-reaction
Disadvantages	Operator- and experience-dependent Strongly calcified plaques with large acoustic reverb artifact Low panoramic view

**Table 3 diagnostics-13-01478-t003:** V-flow and 3D-US advantages and disadvantages for detection of carotid stenosis.

	V-Flow	3D-US
Advantage	Hemodynamic changes, arterial stenosis detection	Accurate evaluation of plaque morphology, surface, and volume
Disadvantage	Operator and experience dependent Low panoramic view	Preliminary results Greater size and weight of the transducer compared to conventional US Operator and experience dependency The need for specialized quantification software to process a 3D image Low panoramic view

## Data Availability

We have done a review of the literature and no specific statement is available.

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
