# Peer review of "New Technologies in the Assessment of Carotid Stenosis: Beyond the Color-Doppler Ultrasound—High Frame Rate Vector-Flow and 3D Arterial Analysis Ultrasound"

_diagnostics, 2023, doi:10.3390/diagnostics13081478_

Round 1

Reviewer 1 Report

Thanks for the opportunity to review the manuscript entitled ”New technologies in the assessment of carotid stenosis. Beyond the color-Doppler ultrasound: high frame rate vector-flow and 3D arterial analysis ultrasound”.

The authours extensively reviewed different carotid artery stenosis diagnostic imaging techniques, from clasic to modern ones, and their impact for a precise and timely diagnosis. These aspects are important for treatment decision, medical or surgical, in order to prevent an atherothrombotic stroke. The advantages and disadvantages of each method were described, together with their availability. It is a well documented and written review.

Author Response

Thanks for your comments; We are proud that you appreciate our work. 

Reviewer 2 Report

The Authors focused on a review of the new technologies in the assessment of carotid stenosis. Beyond the color-Doppler ultrasound: high frame rate vector-flow and 3D arterial analysis ultrasound. The title clearly describes the article. The abstract presents an accurate description of the case and its implications.

In introduction Authors are describe what the hoped to achieve accurately, and clearly state the problem being investigated. I have to say that most of the references are from the last 5 years.

This is an interesting and important topic but I’m not sure if it is systematic and comprehensive review. The manuscript is well structured, but some important facts that the authors should add.  

Key points to consider:

The authors left out the methodology section, which in my opinion is very important for a clear structure of the review.

The methodology section should provide information of literature searching process. If not, is not adequate for review form. Please add details about methodology of this study:

How was the data collected?

What databases the authors used to search for literature?

What were the search guidelines? Keywords? Time of publication?

How many articles were rejected and for what reason?

Please provide flow chart of searching.

I am very impressed with your work in writing this important review.

Author Response

Thanks for your suggestions which helped us to improve the paper.

We made a methodology section, which you suggested us for a clear structure of the review.

The methodology section provide information of literature searching process with details about the methodology of this study and in particolar about data collected, databases used to search for literature, search guidelines, keywords, time of publication. We also provided a flowchart of searching.

Please let us know your opinion.

Round 2

Reviewer 2 Report

The methodology section is good enough. I accept this correction and recommend approval of this paper.